# Closing the gap to effective gene drive in *Aedes aegypti* by exploiting germline regulatory elements

Michelle A. E. Anderson[1,3,4], Estela Gonzalez[1,3,4], Joshua X. D. Ang[1,3,4], Lewis Shackleford[1,3], Katherine Nevard[1,3], Sebald A. N. Verkuijl[1,2], Matthew P. Edgington [1,3], Tim Harvey-Samuel[1] & Luke Alphey[1,3] ✉

CRISPR/Cas9-based homing gene drives have emerged as a potential new approach to mosquito control. While attempts have been made to develop such systems in *Aedes aegypti*, none have been able to match the high drive efficiency observed in *Anopheles* species. Here we generate *Ae. aegypti* transgenic lines expressing Cas9 using germline-specific regulatory elements and assess their ability to bias inheritance of an sgRNA-expressing element (*kmo*[sgRNAs]). Four *shu*-Cas9 and one *sds3*-Cas9 isolines can significantly bias the inheritance of *kmo*[sgRNAs], with *sds3*G1-Cas9 causing the highest average inheritance of ~86% and ~94% from males and females carrying both elements outcrossed to wild-type, respectively. Our mathematical model demonstrates that *sds3*G1-Cas9 could enable the spread of the *kmo*[sgRNAs] element to either reach a higher (by ~15 percentage point) maximum carrier frequency or to achieve similar maximum carrier frequency faster (by 12 generations) when compared to two other established split drive systems.

Advances in insect synthetic biology have allowed the development of a new class of genetics-based pest control technologies, collectively termed 'gene drives'[1]. Gene drives function by pushing a trait of interest, e.g. an allele or other modification, to a higher frequency in a target population than would be predicted solely by the relative fitness of individuals bearing that trait. Such behaviour is useful for pest management as it can potentially be used to spread either a genetic load—leading to target population suppression, or a 'refractory transgene'—reducing the ability of the target population to vector a particular pathogen.

To date, the most widely investigated class of gene drives are the 'homing drives' utilising the targetable endonuclease CRISPR/Cas9. These homing drives function by converting a proportion of the diploid germline cells in heterozygous drive carriers to being homozygous for the drive element[2]. This is achieved by inserting the drive into a particular target locus and designing it such that it includes the necessary single-guide RNA/s (sgRNA/s) to cut the homologue of that

locus on the wild-type chromosome. During the repair of that cut chromosome, the germ-cell may employ the homology-directed repair (HDR) pathway in which it will utilise the drive carrying homologue as a repair template, effectively copying the drive over onto the wild-type chromosome ("homing"). In theory, such behaviour leads to an exponential increase in the proportion of individuals carrying the drive across a targeted population, with the specific dynamics being a function of germline conversion rate, drive fitness costs and the production and fitness of resistant alleles arising through failed conversion attempts.

In addition to the sgRNA arrangement described above, a further requirement for a homing drive is a source of the endonuclease Cas9, which could be located either within the homing locus (autonomous-drive), or at an independent, non-driving, site (split-drive). It is generally assumed that for an efficient drive system, this should be expressed in diploid germline cells during, or immediately prior to, meiosis[3–6]. Double-stranded breaks induced by the expression of

[1]Arthropod Genetics, The Pirbright Institute, Ash Road, Pirbright GU24 0NF, UK. [2]Mathematical Ecology Research Group, Department of Biology, University of Oxford, 11a Mansfield Road, Oxford OX13SZ, UK. [3]Present address: The Department of Biology, University of York, Wentworth Way, York YO10 5DD, UK. [4]These authors contributed equally: Michelle A. E. Anderson, Estela Gonzalez, Joshua X. D. Ang. ✉e-mail: luke.alphey@york.ac.uk

nuclease prior to this period in mitotically active germline cells are thought to be primarily repaired through the non-homologous end joining pathway (NHEJ) or other error-prone repair pathways. Cuts generated after meiosis cannot undergo inter-homologue repair as the homologues have segregated into separate gametes. Cuts during such periods reduce/prevent the chances of successful homing, and dramatically increase the chances of cut-resistant allele formation through mutagenic repair of the double-strand break[4,7]. Much of the iterative process of 'engineering' gene drives has therefore focused on the identification of regulatory elements (i.e. promoters, 3′ UTRs) to increase expression during, and restrict expression to, this relatively narrow window[6–10].

While much of the progress in homing drive refinement has taken place in mosquitoes, this has mainly been confined to *Anopheles gambiae* and *An. stephensi*, where gene drive allele inheritance rates of >95% have been reported even from early prototypes of such drive systems[11,12]. In contrast, while proof of principle homing drives have been demonstrated in the dengue and yellow fever mosquito *Aedes aegypti*, the conversion rates achieved so far have been more modest (highest reported average drive inheritance rate = 80.5% in $w^{U6b-GDe}$;*nup50*-Cas9 trans-heterozygous females, with $w^{U6b-GDe}$;*nup50*-Cas9 trans-heterozygous males showing a lower rate of 66.9%)[10]. While modelling suggests that these rates would be sufficient to underpin gene drive use in field releases, this was based on frequent (10 consecutive weeks), high-rate (2:1 transgenic: WT male population) releases of homozygous males. Subsequent cage population testing of a separate homing drive targeting the kynurenine monooxygenase (*kmo*) allele demonstrated empirically that a split homing drive system could increase in frequency within a target population[13]. When released at a carrier frequency of ~50%, the increase was again relatively modest (maximum frequency 89%) and transient (<6 generations), although in line with modelling predictions. While these studies provide clear proof of principle, substantial efficiency improvements would likely be required before such tools could be practically deployed for mosquito control.

Here we set out to identify and test several regulatory sequences for the expression of Cas9 in *Ae. aegypti* with the aim of improving germline cutting/homing rates to a level more in-line with those observed in the Anopheline mosquitoes. These regulatory sequences included promoters, 5′ and 3′ UTRs from the *Ae. aegypti* homologues of *suppressor of defective silencing 3* (*sds3*), *zero population growth* (*zpg*), *shut-down* (*shu*), *Ewald*, and *nanos* (*nos*), each chosen for their putative germline-biased expression in *Ae. aegypti* or other dipterans. We found that the combination of *sds3*G1-Cas9 and *kmo*sgRNAs provided inheritance rates substantially superior to those previously reported for *Ae. aegypti*, demonstrating that an efficient homing drive system can be built in a seemingly recalcitrant species through extensive testing of regulatory elements and sgRNAs. The characterisation of such tools will aid in the development of improved homing drive systems in *Ae. aegypti* and possibly inform technology development in other Culicine mosquitoes of human or veterinary disease importance.

## Results

### Identification of putative germline expressing regulatory elements and generation of transgenic lines

The precise timing of nuclease activity in the germline has been shown to be key to successful Cas9-based homing drives[7]. For this purpose, we designed constructs with Cas9 under the control of putative promoter fragments derived from *Ae. aegypti* homologues of genes previously identified as having germline expression in *An. gambiae*, *Ae. aegypti* and/or *Drosophila melanogaster*. *Suppressor of defective silencing 3* (*sds3*) encodes a putative component of the histone deacetylase co-repressor complex in *An. gambiae*. Embryonic injection of dsRNA targeting this gene in *An. gambiae* resulted in complete non-formation of the testes and ovaries; injected individuals appeared otherwise

normal, including in their external genitalia[14]. *Zero population growth* (*zpg*) encodes a germline-specific gap junction protein (innexin 4) identified in *D. melanogaster* as playing a role in early germ cell differentiation in both sexes[15]. The mRNA of its homologue was found to be highly expressed in the gonads of *Ae. aegypti*[16]. In *An. gambiae*, the regulatory element of *zpg* was successfully used to express Cas9 and mediated super-Mendelian inheritance (mean >95%) of a gene drive element[17]. In a separate study aimed at developing tools for transgene remobilisation in *Ae. aegypti*, the homologues of the *D. melanogaster* genes *shu*[18], *Ewald*, and *zpg* were shown by RT-PCR, in combination with in situ hybridization, to be expressed in early embryos and ovaries of blood-fed females[19]. The regulatory region of *nanos* (*nos*) has been used extensively in the development of homing drive systems in *D. melanogaster*[20,21] and its *Ae. aegypti* homologue has also been identified and characterised[22]. We performed both 5′ and 3′ RACE on cDNA obtained from dissected testes and ovaries to verify the predicted UTRs of the candidate genes in these organs. A region of approximately 2 kb, which encompassed the entire 5′ UTR of the genes as well as putative promoter regions further upstream, was selected to initiate the expression of Cas9 while the 3′ UTR, including the putative polyadenylation signal, was incorporated into the construct downstream of Cas9 (Supplementary Fig. S1). These constructs were injected into *Ae. aegypti* embryos with a helper plasmid expressing hyperactive piggy-Bac transposase[13] and integration of the construct was identified by screening $G_1$ larvae for mCherry fluorescence. Isolines were subsequently established by individually outcrossing transgenic males from each positive pool. Letters denote the pools the isolines originated from, followed by a number indicating the $G_1$ founder male. For example, isolines *shu*C1 and *shu*C2 were generated from two different $G_1$ males originating from pool C. A total of 15 isolines (Supplementary Table S1) were generated from these crosses.

### Cas9 expressed using *sds3* and *shu* regulatory sequences enable efficient gene drive

We assessed all transgenic isolines for their ability to bias the inheritance of an sgRNA carrying element (*kmo*sgRNAs) described previously[13]. In brief, this construct is inserted within exon 5 of the *kmo* gene and contains four different *Ae. aegypti* RNA pol III promoters, each expressing a different sgRNA targeting the region of *kmo* into which the construct was inserted, together with an AmCyan expression cassette for visual tracking. We performed three versions of crosses described below as $P_{0-2}$ (Paternally inherited nuclease), $M_{0-2}$ (Maternally inherited nuclease), and $I_{0-2}$ (Individual assessments). In the first set of crosses, we generated trans-heterozygous $P_1$ mosquitoes by crossing Cas9 males to *kmo*sgRNAs females, as shown in Fig. 1a, to prevent maternal deposition of Cas9 into embryos which could potentially create NHEJ mutations and affect biasing efficiency[11]. These trans-heterozygous $P_1$ individuals were then crossed to WT of the opposite sex to generate $P_2$ progeny which were screened to determine the inheritance rate of *kmo*sgRNAs (indicated by the presence of the AmCyan marker). $P_1$ male trans-heterozygotes could not be produced from male *shu*B1-Cas9 $P_0$ because *shu*B-Cas9 was found to be linked (as deduced by the absence of Cas9-carrying males at the $P_1$ generation) to the m allele of the M/m sex-determining locus (Fig. 1c; Supplementary Table S5). Larval eye phenotype was also examined as homozygous loss of *kmo* function results in lack of pigmentation in the entirety/patches of the eye (Fig. 1d); manifestation of such phenotypic traits in the $P_2$ larvae would indicate the occurrence of nuclease activity in embryonic somatic cells. If this occurred more in $P_2/M_2$ progeny from female $P_1/M_1$ trans-heterozygotes than male $P_1/M_1$ trans-heterozygotes this would indicate maternal deposition of Cas9/sgRNAs.

We used the *kmo*sgRNAs inheritance rate in $P_1$ progeny from *sds3*G1-Cas9 $P_0$ as the control (48.1%, $n = 441$) for comparison to the various crosses described below as this should indicate normal Mendelian inheritance rates of this element without any drive present in any of

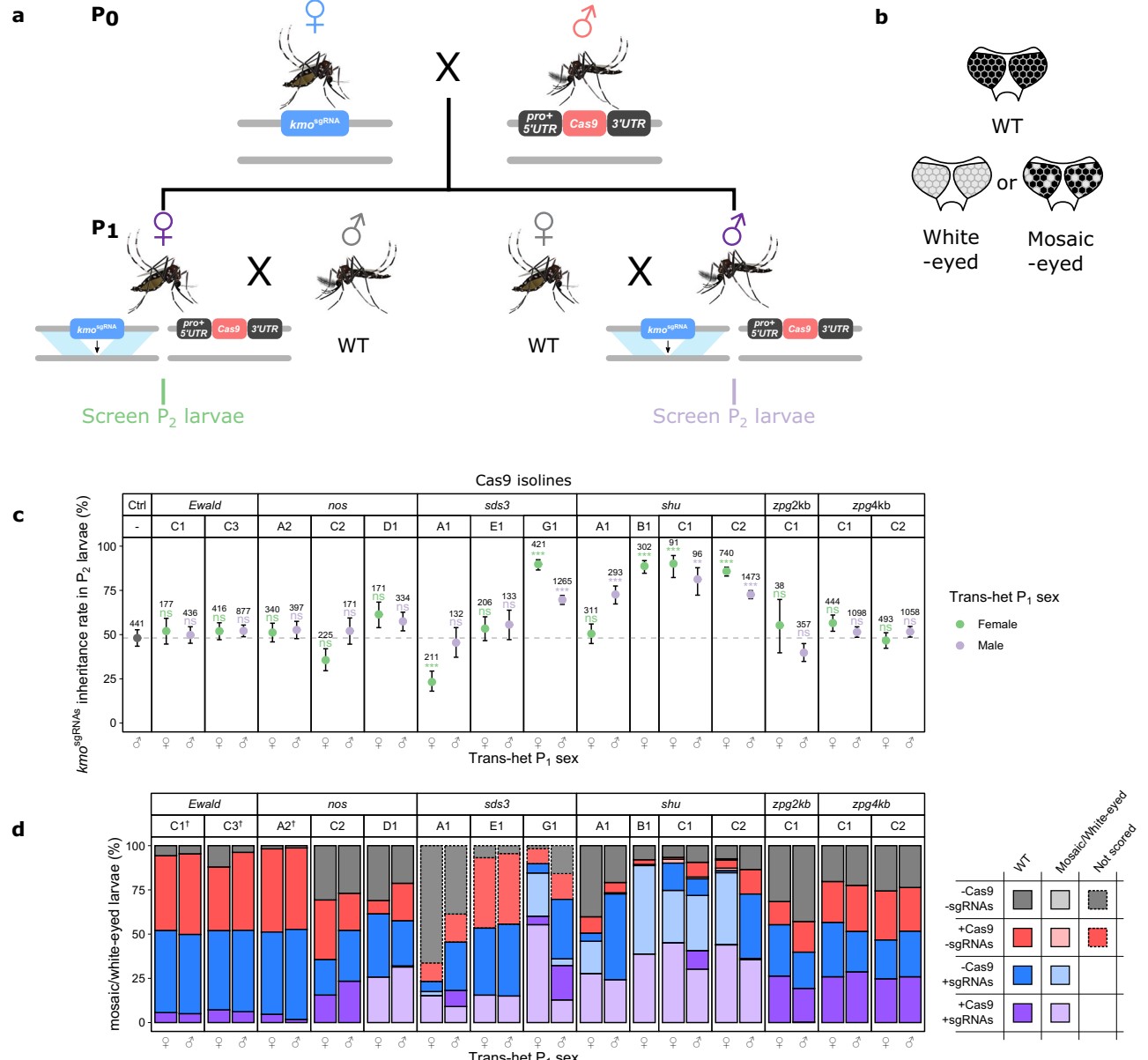

**Fig. 1 | Cas9 expressed by *sds3* and *shu* regulatory elements causes inheritance bias of the *kmo*^sgRNAs element. a** Crossing scheme for determination of Cas9-induced inheritance bias. **b** Illustration of the difference between WT-, mosaic-, or white-eyed phenotype. **c** Inheritance rate of the *kmo*^sgRNAs in P₂ larvae, scored by AmCyan fluorescence. Total number of screened larvae from each cross is presented above corresponding data points. Error bars are the Wilson confidence intervals for the binomial proportion. The confidence intervals are calculated from the pooled progeny count and cannot account for potential over dispersal due to parent by parent 'batch' effects. Statistical significance was estimated using Fisher's two-sided exact test relative to the control inheritance rates represented by the dotted line ($p \geq 0.05$ns, $p < 0.05$*, $p < 0.01$**, and $p < 0.001$***). **d** Percentage of P₂ larvae exhibiting WT or mosaic/white-eye phenotype, according to genotype.†The integration sites for *Ewald*C1-Cas9, *Ewald*C2-Cas9, and *nos*A2-Cas9, isolines are likely linked in trans to *kmo*^sgRNAs indicated by the low representation of the trans-heterozygote and non-transgenic genotypes in P₂. Mosquito figures obtained from Ramirez[38,39]. Source data are provided as a Source data file. pro = promoter, ♂ = male, ♀ = female.

their P₀ parents. P₁ crosses with four *shu*-Cas9 (72.6–90.1%, apart from the P₁ cross with *kmo*^sgRNAs;*shu*A1-Cas9 trans-heterozygous females which was not significant) and one *sds3*-Cas9 (69.6% and 89.8%) isolines caused significant super-Mendelian inheritance of the *kmo*^sgRNAs element in the P₂ larvae screened (Fig. 1c; Supplementary File 1). Inheritance rate of the *kmo*^sgRNAs element was not significantly different (Fisher's two-sided exact test, $p > 0.059$) from the control in the crosses with the remaining Cas9 isolines, except for *kmo*^sgRNAs;*sds3*A1-Cas9 P₁ trans-heterozygous females where significant reduction of the inheritance rate was observed in their P₂ progeny. As *zpg*-Cas9 has been shown in *An. gambiae*[17] to be highly successful in causing biased

inheritance, a longer promoter region (~4.0 kb) was cloned to create *zpg*4kb-Cas9 after observing that *zpg*2kb-Cas9 did not induce inheritance bias. However, this longer version also failed to significantly bias the inheritance of *kmo*^sgRNAs in the P₂ larvae (Fig. 1c).

Mosaic- or white-eyed P₂ larvae were almost completely absent in all genotypes when Cas9 was expressed by *Ewald*, *zpg*2kb, and *zpg*4kb (Fig. 1d; Supplementary File 1). This indicates that Cas9 was either produced at relatively low levels or not expressed at all in these lines. Nuclease activity as evidenced by mosaic or white eyes could be detected in several other lines although they did not cause a significant increase in the inheritance of *kmo*^sgRNAs (Fig. 1c, d). P₂ larvae with

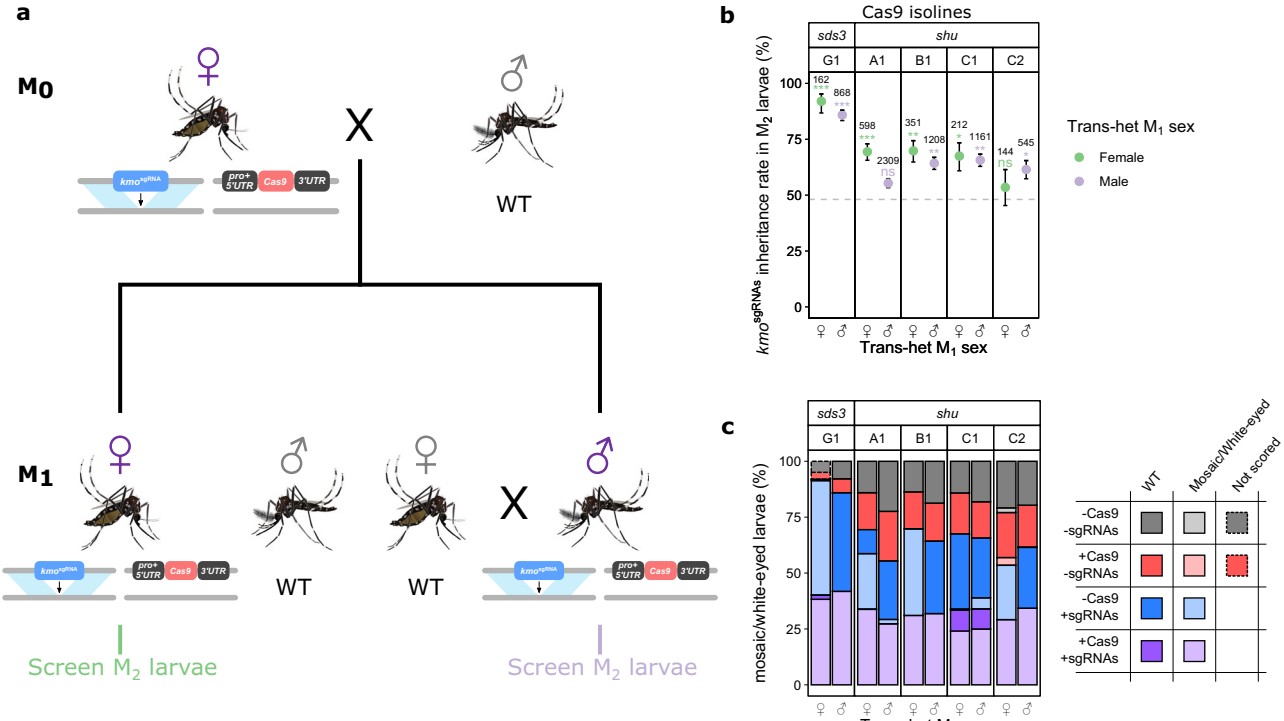

**Fig. 2 | Inheritance bias of the *kmo*^sgRNAs element could still occur with maternal deposition of Cas9 and/or sgRNAs. a** Crossing scheme for determination of Cas9-induced inheritance bias. **b** Inheritance rate of the *kmo*^sgRNAs in $P_2$ larvae, scored by AmCyan fluorescence. Total number of screened larvae from each cross is presented above corresponding data points. Error bars are the Wilson confidence intervals for the binomial proportion. The confidence intervals are calculated from the pooled progeny count and cannot account for potential over dispersal due to parent by parent 'batch' effects. Statistical significance was estimated using Fisher's two-sided exact test relative to the control inheritance rates represented by the dotted line ($p \geq 0.05$^ns, $p < 0.05$*, $p < 0.01$**, and $p < 0.001$***). **c** Percentage of $P_2$ larvae exhibiting WT or mosaic/white-eye phenotype, according to genotype. Mosquito figures obtained from Ramirez[38,39]. Source data are provided as a Source data file. pro = promoter, ♂ = male, ♀ = female.

mosaic/white eyes were observed in individuals inheriting both transgenes from *kmo*^sgRNAs;*nos*D1-Cas9, *kmo*^sgRNAs;*sds3*A1-Cas9, and *kmo*^sgRNAs;*sds3*E1-Cas9 trans-heterozygous parents, indicating nuclease expression and activity in those trans-heterozygotes (Fig. 1c). Mosaic/white eyes were further observed in 29.4% of the *kmo*^sgRNAs-only progeny of *kmo*^sgRNAs;*sds3*A1-Cas9 trans-heterozygous mothers, indicating maternal deposition of at least Cas9 into this progeny. This cross also happens to be the only cross with a significant reduction (Fisher's two-sided exact test, $p < 0.0001$) of the *kmo*^sgRNAs inheritance rate. However, no further investigations were carried out on this isoline as it did not increase the inheritance of the *kmo*^sgRNAs element.

### Parental sex affects biasing efficiency and mosaicism

The five inheritance-biasing Cas9 isolines were further assessed to determine biasing efficiency of trans-heterozygotes when both the Cas9 and sgRNA are inherited maternally, thereby maximising deposition of Cas9 and/or sgRNA and potential formation of resistant alleles (Fig. 2). For this assessment, female $M_0$ trans-heterozygotes were crossed to WT males to generate trans-heterozygous $M_1$ individuals. These $M_1$ adults were then reciprocally crossed to WT of the opposite sex and their progeny ($M_2$) screened for inheritance of the *kmo*^sgRNAs transgene. Comparing the range of *kmo*^sgRNAs inheritance rates among $P_2$ and $M_2$ across all the four versions of crosses (Figs. 1c and 2b), the *sds3*G1-Cas9 isoline (69.6−92.0%) was found to have caused the highest inheritance bias, followed by *shu*C1-Cas9 (65.6−90.1%), *shu*B1-Cas9 (64.2−88.7%), *shu*C2-Cas9 (61.5−85.8%), and *shu*A1-Cas9 (69.4−72.7%).

We investigated the possible effects grandparental sexes at the $P_0/M_0$ levels may have on biasing efficiencies of $P_1$ adults by comparing biasing rates of $P_1$ males/females to their $M_1$ males/females

equivalents, respectively. Any difference found between the pairs would have been caused by the $P_0/M_0$ sexes as these are the only variable in these pairs of crosses. We found grandparental sex to significantly (Fisher's two-sided exact test, $p < 0.0016$) affect the ability of the $P_1/M_1$ trans-heterozygotes to cause inheritance bias in all crosses except in the *kmo*^sgRNAs;*sds3*G1-Cas9 $P_1/M_1$ female trans-heterozygotes where no significant differences in biasing efficiency was detected (Fisher's two-sided exact test, $p = 0.53$).

All *shu*-Cas9 isolines exhibited somatic/zygotic expression, as evidenced by high mosaicism rates in trans-heterozygous $P_2/M_2$ larvae regardless of their parental sexes. Exceedingly high levels (>98.0%) of mosaicism were observed with *kmo*^sgRNAs;*shu*A1-Cas9, *kmo*^sgRNAs;*shu*B1-Cas9, and *kmo*^sgRNAs;*shu*C2-Cas9 $P_1/M_1$ trans-heterozygotes (Figs. 1d and 2c). Mosaicism rates in the $P_2/M_2$ larvae which originated from *kmo*^sgRNAs;*shu*C1-Cas9 $P_1/M_1$ trans-heterozygotes were unexpectedly affected by the sex of the $P_0/M_0$ grandparents, but not $P_1/M_1$ parents. In these crosses, mosaicism rates in the $P_2/M_2$ larvae, especially those that inherited only *kmo*^sgRNAs, were much higher when Cas9 originated from a $P_0$ male (mosaicism rate in *kmo*^sgRNAs-only progeny from $P_1$ males: 76.9%; $P_1$ females: 65.9%) rather than an $M_0$ female (mosaicism rate in *kmo*^sgRNAs-only progeny from $P_1$ males: 15.8%; $P_1$ females: 1.4%). Due to its consistent and high levels of inheritance bias in all directions of crosses, we further investigated the drive dynamics in *kmo*^sgRNAs;*sds3*G1-Cas9 trans-heterozygotes.

### *sds3*G1-Cas9 induces >95% inheritance rate of *kmo*^sgRNAs in both trans-heterozygote sexes

Having established *sds3*G1-Cas9 as being able to cause substantially increased inheritance of *kmo*^sgRNAs, we then examined this at the individual-level to determine the level of variance between

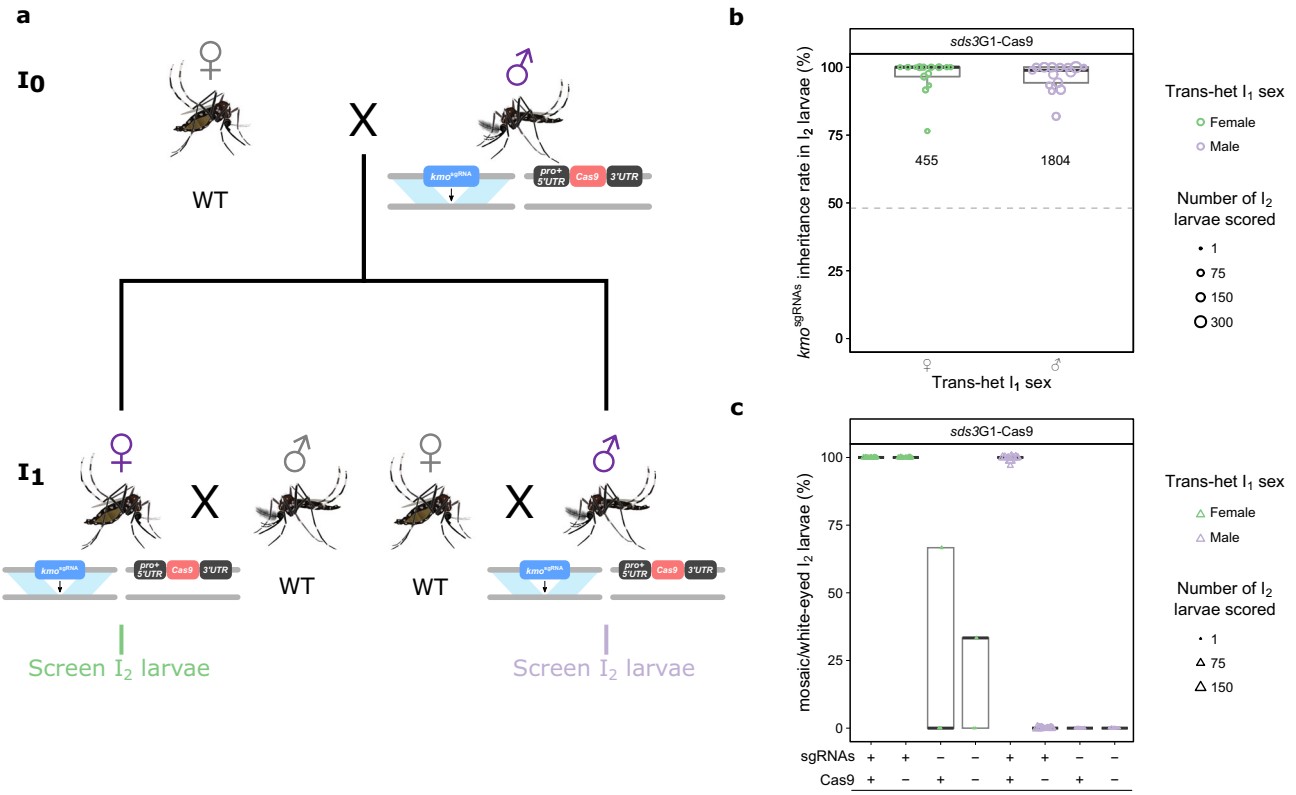

**Fig. 3 | Drive characteristics of individual $kmo^{sgRNAs}$;$sds3$G1-Cas9 trans-heterozygotes. a** Crossing scheme for determination of $sds3$G1-Cas9-induced inheritance bias. **b**, **c** Boxplots of $kmo^{sgRNAs}$ inheritance (**b**) and eye phenotype (**c**) rates of $I_2$ larvae. The median value is indicated by a thick black line and the maximum and minimum ends of whiskers represent the most extreme data point that is no more than 1.5 times the interquartile range. The size of each symbol is scaled to the amount of $I_2$ progeny scored. Total number of screened larvae from each cross is presented below corresponding data points. Mosquito figures obtained from Ramirez[38,39]. Source data are provided as a Source data file. pro = promoter, ♂ = male, ♀ = female.

trans-heterozygous individuals. As such, we repeated the previous crosses but initiated them with trans-heterozygous $I_0$ adults (Fig. 3a) and tracked the inheritance of transgenes and eye phenotypes of the $I_2$ larvae produced by each individual $I_1$ trans-heterozygote. An average of 97.0% (95% CI: 96.1–97.7%; $n = 30$ $I_1$ females, 1804 larvae) and 97.6% (95% CI: 95.7–98.6%; $n = 14$ $I_1$ females, 455 larvae) of $I_2$ progeny were found to have inherited the $kmo^{sgRNAs}$ from male and female $I_1$ trans-heterozygotes, respectively (Fig. 3b). Mosaic- or white-eyed phenotypes were detected in more than 99.6% of the $I_2$ larvae carrying both transgenes regardless of the sex of their $I_1$ parents. In $I_2$ larvae which inherited only $kmo^{sgRNAs}$, the average rate of such phenotypes was found to be 100% if they descended from a female $I_1$ or 0% if they descended from a male $I_1$. Additionally, mosaicism was only observed at a relatively low rate (27.3%, 3/11 larvae) among non-$kmo^{sgRNAs}$ inheriting $I_2$ larvae descended from female $I_1$ trans-heterozygotes and 0% (0/55 larvae) when descended from male $I_1$ trans-heterozygous parents (Fig. 3c). Taken together, these observations suggest that sgRNAs and/ or Cas9 were deposited by female trans-heterozygotes but not male trans-heterozygotes and that somatic nuclease activity occurred in individuals carrying both transgenes even in the absence of deposition.

**Model predicts $sds3$G1-Cas9 to enable faster spread than other split drives in Ae. aegypti**

The work above has given clear indications that a split drive based on $sds3$G1-Cas9 could outperform existing systems[13,23]. Thus, we wished to approximate the degree of improvement expected from employing $sds3$G1-Cas9 as opposed to the most viable alternatives in previous Ae. aegypti experimental work, namely $kmo^{sgRNAs}$;$bgcn$-Cas9[13] and $w^{U6b-GDe}$; $nup50$-Cas9[10]. We use a model analogous to that from our split drive

cage trial study[13] (Supplementary Note 1; Supplementary Tables S2–4 and S7) and, to ensure fairness, consider the same cage trial setup used previously—i.e. 50% trans-heterozygous females and 50% wild-type males. Parameter values for the $bgcn$-Cas9 split drive are taken from our previous work—i.e. $bgcn$-Cas9 and $kmo^{sgRNAs}$ elements produce 21% and 19% homozygote fitness costs, respectively while heterozygotes display approximately wild-type fitness. Inheritance rates of $kmo^{sgRNAs}$ from trans-heterozygous $bgcn$-Cas9;$kmo^{sgRNAs}$ males and females were observed to be 68% and 77%, respectively. Inheritance rate and fertility parameters for $nup50$-Cas9 and $w^{U6b-GDe}$ are drawn directly from the mathematical modelling in Li et al.[10]. Parameter values derived for $sds3$G1-Cas9 are calculated from simple heterozygote and homozygote viability assays (Supplementary Tables S2, S3). Here $sds3$G1-Cas9 is found to produce approximately wild-type fitness in heterozygotes of both sexes. Based on mendelian inheritance rates and the number of WT mosquitoes obtained in the homozygous viability cross we estimated fitness costs of ~55% in homozygous males and ~9% in homozygous females. We assume no change in the fitness costs associated with $kmo^{sgRNAs}$. The average inheritance rates of $kmo^{sgRNAs}$ from $kmo^{sgRNAs}$;$sds3$G1-Cas9 trans-heterozygous males and females crossed to WT mosquitoes are found to be ~86% ($n = 3937$ larvae) and ~94% ($n = 1038$ larvae), respectively by combining the data from all such crosses performed above (Supplementary Table S4).

These results clearly demonstrate that the improved efficiency of the $sds3$G1-Cas9 element studied here could translate into an improvement in invasiveness of split drives when compared to a previous cage trial study using $bgcn$-Cas9 (Fig. 4). In particular, for the modelled release scenario, $sds3$G1-Cas9 can produce a maximum $kmo^{sgRNAs}$ carrier percentage of ~95%, whereas the $bgcn$-Cas9 equivalent reaches only ~80%. Perhaps more importantly, the $sds3$G1-Cas9 version

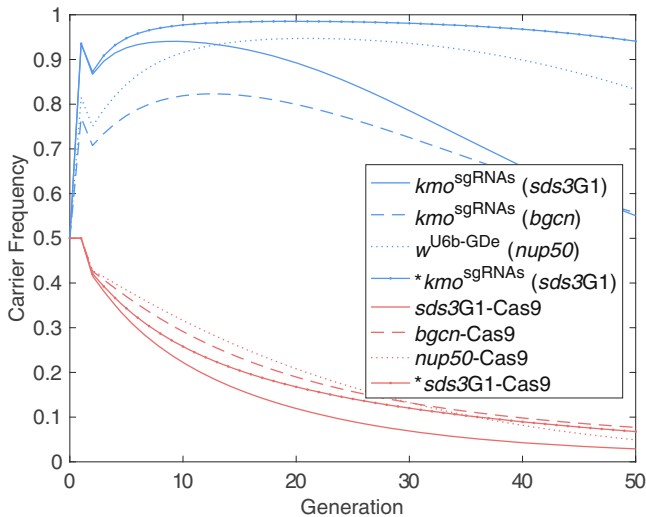

**Fig. 4 | Model predicted split drive dynamics using *sds3*G1-Cas9, *nup50*-Cas9 and *bgcn*-Cas9.** Red lines represent carrier frequencies for the Cas9 elements, while blue lines represent the resulting carrier frequencies of the *kmo*^sgRNAs or *w*^U6b-GDe elements. Here, results for *kmo*^sgRNAs;*bgcn*-Cas9 (dashed lines) are identical to those previously reported and the computational simulations were shown to approximate the experimental cage data[13]. Results for *nup50*-Cas9;*w*^U6b-GDe (dotted line) are based on parameter values listed in Li et al.[10]. and *kmo*^sgRNAs;*sds3*G1-Cas9 (solid lines) are predictions based on experimental data generated here. Solid lines marked with dots display hypothetical results for a *kmo*^sgRNAs;*sds3*G1-Cas9 system in which *kmo*^sgRNAs fitness costs are reduced to levels observed for *w*^U6b-GDe (marked with * in the figure legend). Parameter values for all scenarios are detailed in Supplementary Note 1.

can maintain a carrier percentage >85% for 23 generations (~2 years, given ~12 generations per year which is realistic in tropical regions)[24,25]. Note that here we consider a carrier percentage of >85% to be a reasonable target for split drives carrying efficient refractory elements for *Ae. aegypti* transmitted pathogens (e.g. dengue, Zika, and Chikungunya viruses), based on typical basic reproduction number ($R_0$) estimates that are in the range $R_0 = 1-4.5$[26-34]. In the longer term, *kmo*^sgRNAs carrier percentages with *bgcn*-Cas9 are predicted to overtake those for the *sds3*G1-Cas9 version due to lower overall fitness costs. However, by this point frequencies are in decline and the peak efficacy would have passed more than two years (i.e. >24 generations) prior.

A different overall pattern is observed upon comparison between *kmo*^sgRNAs;*sds3*G1-Cas9 and the *w*^U6b-GDe;*nup50*-Cas9 design of Li et al.[10]. Specifically, while each system is expected to attain almost identical maximum carrier percentages (~95%), the two approaches display very different temporal dynamics. For instance, the *kmo*^sgRNAs;*sds3*G1-Cas9 system reaches its maximum carrier percentage more quickly (9 generations) than *w*^U6b-GDe;*nup50*-Cas9 (21 generations). This is due to the increased rate of inheritance observed for *kmo*^sgRNAs;*sds3*G1-Cas9 compared to *w*^U6b-GDe;*nup50*-Cas9. However, while it may take longer to spread through a target population, the *w*^U6b-GDe;*nup50*-Cas9 system is able to maintain a high carrier frequency for longer. For example, *w*^U6b-GDe;*nup50*-Cas9 is predicted to maintain a carrier percentage of >85% for 41 generations which is 18 generations (or ~1.5 years) longer than *kmo*^sgRNAs;*sds3*G1-Cas9. This is due to the greater homozygote fitness costs of *kmo*^sgRNAs compared to the *w*^U6b-GDe element.

The above results clearly demonstrate the potential of *kmo*^sgRNAs;*sds3*G1-Cas9, to outperform the two previous best examples in *Ae. aegypti* in all but one aspect—its ability to persist at high frequency. This was only bettered by *w*^U6b-GDe;*nup50*-Cas9 due to the lower fitness costs of the target *w*^U6b-GDe element. Given that fitness of the *kmo*^sgRNAs element could be potentially improved by providing a recoded rescue[35], we go on to model a hypothetical scenario in which the

*kmo*^sgRNAs element has reduced fitness costs, in line with those for *w*^U6b-GDe (Fig. 4). Model predictions indicate that, in this scenario, *kmo*^sgRNAs will undergo a rapid increase in frequency due to the high inheritance rates associated with *sds3*G1-Cas9. However, now the *kmo*^sgRNAs;*sds3*G1-Cas9 system with reduced fitness costs is predicted to reach a greater maximum carrier percentage (~98.5%) and persist at >85% carrier percentage for longer (66 generations) than the *w*^U6b-GDe;*nup50*-Cas9 system (predicted maximum carrier percentage ~94.7% and 41 generations at >85% carrier percentage). This would give an increase in the period of high efficacy of 25 generations (or ~2 years).

## Discussion

In this study we assessed Cas9 expressing lines utilising six different promoter/3′ UTR combinations and found 1/3 *sds3*-Cas9 and 4/4 *shu*-Cas9 isolines to be capable of significantly increasing inheritance of our sgRNA-expressing element (*kmo*^sgRNAs). This finding was despite most of the genes associated with the six regulatory elements having been shown to be expressed in the ovaries of adult *Ae. aegypti*, with the exception of *sds3*, where its *An. gambiae* orthologue is thought to be essential in gonad development of both sexes. The *nos* and *zpg* regulatory elements have been used in *Ae. aegypti* in an autonomous homing-based gene drive system and shown to have caused average drive element inheritance rates of <75%[36]. The lack of any indication of Cas9 activities from *Ewald*-Cas9 and *zpg*-Cas9 in the present study may be due to the putative regulatory elements not encompassing the regions necessary for transcription and/or translation of Cas9. *zpg* was recently used to express Cas9 in a single locus gene drive in *Ae. aegypti*[36]. They worked in the Higgs white eye strain using a 1.7 kb promoter fragment roughly similar to our initial 2 kb fragment (although they note the promoter as having a 144-bp deletion which was 825 bp upstream of the +1ATG of *zpg*), and a 1.3 kb 3'UTR and so there are differences to our regulatory elements. They observed a moderate inheritance bias (66%) in females when they targeted Carb109. In light of their results, *zpg* could be further explored for use in expressing Cas9, perhaps in loci previously characterised as permissive to germline expression. Data collected on eye phenotypes appears to indicate that Cas9 was transcribed and translated from at least one of each of the *nos*-Cas9, *sds3*-Cas9, and *shu*-Cas9 isolines and that integration sites of the transgenes played a significant role in transgene expression.

Only three other studies have described the development of split homing-based drives in *Ae. aegypti*[10,13,23]. In the first of those studies, a total of five Cas9-expressing lines[10] were tested for their abilities to drive an sgRNA element (termed U6b-GDe) inserted into the *white* (*w*) gene which also causes white-eye phenotype when both copies of the gene are disrupted. There, nuclease activity was detected in all five lines but biased inheritance was only observed in the *exu*-Cas9 and *nup50*-Cas9 strains. Inheritance rates of the *w*^U6b-GDe were shown to be ~50% and ~71% from the trans-heterozygous males and females with *exu*-Cas9 and ~66.9% and ~80.5% from their equivalents with *nup50*-Cas9. The second study described the use of a Cas9-expressing line (*bgcn*-Cas9) to drive the *kmo*^sgRNAs element also used in the present study[13]. Results from three replicate crosses showed the inheritance of the *kmo*^sgRNAs element to be increased to between 50.8–68.1% and 75.5–78.2% among the progeny originating from trans-heterozygous fathers and mothers, respectively. It is worth noting that inheritance bias was shown to be stronger from the trans-heterozygous mothers compared to fathers in both studies.

The third study investigated the efficacy of *bgcn*D-Cas9 from Anderson et al.[13], *sds3*G1-Cas9 (also used in the present study) from Verkuijl et al. (2021)[23], and *nup50*-Cas9 from Li et al.[37] to drive the inheritance of the *w*^U6b-GDe element from Li et al.[10]. The *sds3*G1-Cas9 was tested under conditions similar to the M and P crosses in the present study, but a marginally significant inheritance bias (67%, $n = 176$, Fisher's two-sided exact test, $p = 0.0495$) of the *w*^U6b-GDe element was

shown only in one out of the four crossing regimes ($F_0$ Cas9-bearing female, $F_1$ trans-heterozygous female), whereas the same Cas9 isoline mediated significant bias of the $kmo^{sgRNAs}$ inheritance in all six combinations of parental crossing in our hands (Figs. 1–3). This suggests that the efficacy of a split drive does not depend exclusively on the spatiotemporal control of the Cas9 nuclease but also on the regulation of sgRNA expression and/or the chromatin context within which the sgRNA element is inserted in the genome. It is therefore imperative in future split drive development for researchers to focus on creating a compatible pair of elements with high drive efficacy rather than optimising the two elements separately.

Although nuclease activity in the sds3G1-Cas9 and all shu-Cas9 lines were not completely restricted to the germline—the sds3G1-Cas9 line has demonstrated the highest average inheritance biasing efficiency while the shu regulatory region utilised here was found to be the most robust to positional effects in causing inheritance bias in Ae. aegypti to date. Positional effects are well documented with transgenes where the chromosomal context in which a transgene is inserted may affect its relative expression levels. We hypothesised that insertions into intergenic regions may be more likely to be silenced than those loci close to or within genes. Nearby enhancers may also benefit transgene cassettes and upregulate their expression. We did not observe any such pattern however, with three out of four shu-Cas9 lines inserted within a gene and one being intergenic (Supplementary Table S5), and all four able to bias the inheritance of $kmo^{sgRNAs}$. The most active sds3G1-Cas9 line was intragenic however, so was line sds3E-Cas9, with the third insertion being intergenic. While it may be more likely to achieve robust expression by inserting a transgene in or near another gene, the germline specific expression pattern desired here may be more sensitive to nearby enhancers/silencers than other transgenic cargo. Cas9 was also shown to be expressed somatically and/or deposited maternally, both of which may confer fitness costs (especially when sgRNAs target an essential gene) and/or contribute to resistance formation[4,7,12]. Despite this potential for generation of resistant alleles, sds3G1-Cas9 achieved the highest rate of inheritance when inherited maternally. We did however observe a decrease in the capacity to bias the inheritance of $kmo^{sgRNAs}$ by shu-Cas9 when inherited maternally.

We have shown that a near 100% germline cutting and inheritance biasing rates are achievable in Ae. aegypti. Taken together with the evident improvement in modelled outcomes, our study demonstrates the feasibility of substantially enhancing drive efficiency at the individual level and invasiveness at the population level by optimising Cas9 expression with different promoters and associated regulatory elements. This suggests Ae. aegypti is not recalcitrant to Cas9-based homing gene drives and that further refinements to drive components can lead to useful drive systems in this species.

## Methods

### Plasmids and cloning
Total RNA was extracted from testes and ovaries dissected from 5–7 days post eclosion Liverpool adults using Trizol (Life Technologies) according to the manufacturer's instructions. RACE ready cDNA was prepared and 5′ and 3′ RACE PCRs were performed using the SMARTer 5′ and 3′ RACE kit (Takara 634858) according to the manufacturer's instructions. Amplicons were purified using the NucleoSpin Gel and PCR Clean-up kit (Machery Nagel 740609.250) and cloned using the CloneJET PCR Cloning Kit (Thermo Scientific K1232) then Sanger sequenced. Primers are listed in Supplementary Table S6.

For sds3 and zpg2kb constructs, primers listed in Supplementary Table S6 were used to amplify the promoter and 3′ UTR fragments from Ae. aegypti Liverpool strain genomic DNA extracted using the NucleoSpin Tissue DNA extraction kit (Machery-Nagel 740952.250). Amplicons were visualised by gel electrophoresis and purified using

the NucleoSpin Gel and PCR clean up kit (Machery-Nagel 740609.250). Promoter fragments were digested with NotI/XhoI and ligated to AGG1207[13] digested with NotI/XhoI. In a sequential step the intermediate plasmids were digested with PacI and ligated to the 3′ UTR amplicons digested with PacI/AsiSI.

Shu, nos, and Ewald promoters were cloned as above, however the intermediate plasmids were digested with FseI/AscI to remove the T2A-GFP-P10 3′ UTR and this was replaced with the native 3′ UTR amplified and purified as above.

zpg4kb was cloned by HiFi, in a two-step process. Initially a 2 kb promoter fragment and the 3′ UTR were cloned using the same procedure as shu, nos and Ewald. This 2 kb promoter fragment was then removed by digesting with NotI/XhoI, and a 4 kb promoter fragment was amplified from Liverpool gDNA using the primers listed in Supplementary Table S6. These two fragments were then assembled using the NEBuilder HiFi DNA Assembly Master Mix (New England Biolabs E2621L), according to the manufacturer's instructions.

All plasmids were prepared for microinjection using the Nucleo-Bond Xtra Midiprep kit EF (Machery-Nagel 740410.50) and confirmed by Sanger sequencing. Constructs are depicted in Supplementary Fig. S1 and complete plasmid sequences are available from NCBI accession numbers: sds3-Cas9 OP823141, zpg2kb zpg2kbCas9 OP823142, nos-Cas9 OP823143, Ewald-Cas9 OP823144, zpg4kb-Cas9 OP823145, shu-Cas9 OP823146.

### Mosquito rearing
No ethical approval was required for working with invertebrate species; however all work was approved by the BAGSMC at The Pirbright Institute. Aedes aegypti (Liverpool wild-type (WT) and transgenic strains) were maintained as previously described[13]. Briefly, insects were housed in an insectary at 28 °C and 75% RH with a 14/10 day/night light cycle. Larvae were reared in purified water and fed on ground TetraMin flake fish food (TetraMin 769939). Adults were fed 10% sucrose ad libitum, and bloodfed on defibrinated horse blood (TCS HB030) using a Hemotek (Hemotek, Inc AS6W1-3) membrane feeder covered with Parafilm (Bemis HS234526B).

### Generation of Cas9-expressing Ae. aegypti
Embryonic microinjections were performed as described previously[13]. Briefly, 1–2 h embryos were collected and manually aligned using a fine paint brush. Lines of ~100 embryos were adhered to a plastic coverslip with double-sided tape, allowed to desiccate slightly and then covered with halocarbon oil 27 (Sigma H8773). Embryos were injected using Quartz capillaries (Sutter QF1007010) pulled into very fine needles with a Sutter Instruments P2000 laser pipette puller. Injection mixes contained 500 ng/μl Cas9 expression construct and 300 ng/μl AGG1245 AePUb-hyperactive piggyBac[13]. $G_0$ survivors were reared to adulthood as described above. $G_0$ males were crossed individually to 5 WT virgin females for at least 2 days before being combined into cages of approximately 20 $G_0$ males and the 100 WT females they mated. Twenty $G_0$ females were crossed to WT males in pools at a ratio of 1:1. After blood feeding, $G_1$ eggs were collected and hatched under vacuum for synchronized hatching and screened at the L3–L4 larvae stages. Screening for fluorescence was performed on a Leica MZ165C fluorescence microscope and the appropriate filter set (AmCyan or mCherry).

### Crosses for drive assessment
Detailed crossing schemes can be found in Figs. 1a, 2a, and 3a for the different assays carried out. Briefly, >10 adult transgenic mosquitoes between 3 and 7 days post eclosion were crossed to >10 WT or $kmo^{-/-}$ adults of the opposite sex in 15 x 15 x 15cm cages (Bugdorm 4S1515). These were bloodfed and eggs collected, hatched, and screened as L3–L4 larvae under a Leica MZ165C fluorescence microscope for the presence of the marker and eye phenotype.

## Statistics and reproducibility

A basic power analysis was performed to assist in determining the sample size. For 0.8 power it was determined that at least 200 samples should be analysed to detect a 10% difference in inheritance rates. Crosses were performed as described as above and the number of progeny screened is presented in each figure above the data point. Larvae/pupae were screened and separated based on genotype (indicated by the presence of fluorescent markers) then individuals were randomly selected for experiments from this pool. No data were excluded from the analyses. The investigators were not blinded to allocation during experiments and outcome assessment.

## Adapter ligation-mediated PCR for insertion confirmation of *sds3*G-Cas9 and *shu*-Cas9 isolines

Genomic DNA from pools of >10 individuals of *sds3*-Cas9 and all *shu*-Cas9 isolines was extracted using the NucleoSpin Tissue genomic DNA extraction kit (Machery-Nagel). Genomic DNA was digested with either *Bam*HI-HF (NEB R3136), *Msp*I (NEB R0106), or *Nco*I-HF (NEB R3193). Adapters (Supplementary Table S6) were ligated with T4 DNA ligase (NEB M0202) overnight at 14 °C. Primary and nested PCRs were carried out using DreamTaq polymerase (Thermo Fisher Scientific EP0712) using the primers listed in Supplementary Table S6. Amplicons were purified using NucleoSpin Gel and PCR Clean-up Kit and Sanger sequenced using the primers listed in Supplementary Table S6. Genomic locations are listed in Supplementary Table S5.

## Mathematical modelling

In a previous study we utilised a mathematical model near identical to that used here[13]. The single difference between models is that here we allow different relative fitness parameters for males and females of each genotype. The model structure is described further in the Supplementary Note 1. The associated parameter values used to compare the performance of *sds3*-Cas9 and *bgcn*-Cas9 based split drive systems are listed in Supplementary Table S7.

## Reporting summary

Further information on research design is available in the Nature Portfolio Reporting Summary linked to this article.

# Data availability

The data generated or analysed in this study are provided in the Supplementary Information/Source data file. Source data are provided with this paper.

# Code availability

R version 4.0.2 was used for statistical analysis. All code used in the analysis of this manuscript is available at: https://doi.org/10.15124/15640775-cffd-40e3-9055-890870d15db9.

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

## Acknowledgements

M.A.E.A., E.G., J.X.D.A., L.S., K.N., and L.A. were funded through a Defense Advanced Research Projects Agency (DARPA) award [N66001-17-2-4054] to Kevin Esvelt at MIT. M.P.E. was supported by the Wellcome Trust [110117/Z/15/Z]. For the purpose of Open Access, the author has applied a CC BY public copyright licence to any Author Accepted Manuscript (AAM) version arising from this submission. L.A. and T.H.S. were funded by the UK Biotechnology and Biological Sciences Research Council [BBS/E/I/00007033, BBS/E/I/00007038, and BBS/E/I/00007039 strategic funding to The Pirbright Institute]. S.A.N.V. was supported by the UK Biotechnology and Biological Sciences Research Council [BB/M011224/1]. The views, opinions and/or findings expressed are those of the authors and should not be interpreted as representing the official views or policies of the U.S. Government. The funders had no role in study design, data collection and analysis, decision to publish, or preparation of the manuscript.

## Author contributions

M.A.E.A., T.H.S., and L.A. conceived and designed the experiments. M.A.E.A., E.G., J.A., S.A.N.V., L.S., and K.N. performed the experiments, collected, and analysed the data. M.P.E. performed mathematical modelling. J.A. prepared the initial draft of the manuscript. All authors provided comments/edits and approved the final draft for submission.

## Competing interests

The authors declare no competing interests.
