## [Peer Review File · Nature Communications]

Reviewers' Comments:

Reviewer #1:

Remarks to the Author:

Nature Communications Review:

In their manuscript entitled "Novel germline regulatory elements enable highly efficient gene drive in the yellow fever mosquito *Aedes aegypti*," Anderson and colleagues describe novel transgenic strains of *Aedes aegypti* that utilize germline regulatory elements to drive Cas9 expression. They demonstrate the efficacy of these lines to bias the inheritance of a kmo-sgRNA construct as a proof-of-principle for an efficient gene drive of the 'split drive' style.

Gene drives (and CRISPR/Cas9 followed by HDR in general) in *Aedes aegypti* has until now underperformed as compared to *Anopheles gambiae*. With these new lines, Anderson and colleagues have helped to close that gap and their work will be of substantial interest to a wide variety of mosquito biologists, including those interested in developing gene drives for vector control as well as those using CRISPR/Cas9 and HDR to understand basic mosquito biology. In addition to establishing these new lines as viable drivers for gene drive purposes, they use carefully tracked crossing experiments to demonstrate convincingly that maternal inheritance of Cas9 can contribute functionally to inheritance biasing in these schemes. Finally, having settled on one line (SDS3G1-Cas9) that had high levels of inheritance bias in multiple crossing schemes, they evaluated crosses derived from individual animals to determine the levels of inter-individual variability. In all, these data are convincing and experiments were well controlled (but see comments below).

Major comments:

1) There is a marked difference in activity among different isolines. In addition to the linkage to the m locus for one isolate of *sds3*, is there any evidence of where the other insertions lie? Approaches such as splinkerette PCR or TagMapping can be used to map piggyBac insertions and knowing where these lines lie could be very useful for future work in mosquito transgenesis. This is especially important as the point raised in the discussion acknowledges: "that integration sites... played a significant role in transgene expression"

Minor comments:

2) I found Figure 1d and 2c very difficult to interpret at first, especially the differences between the WT and not scored categories in the top two rows. Could a pattern be used instead of different border colours?

3) The graphic design of Figure 2 was initially confusing to me as I mistook panel 2b as referring to the left arm of the cross depicted in 2a (and vice-versa for 2c). These panels could be potentially re-arranged to avoid that confusion?

Reviewer #2:

Remarks to the Author:

The manuscript "Novel germline regulatory elements enable highly efficient gene drive in the yellow fever mosquito *Aedes aegypti*" by Anderson and colleagues describes the ability of five different regulatory sequences to express Cas9 and bias the inheritance of the gRNA construct in a context of split gene drive in *Aedes aegypti*. The promoter and 3'UTR of those five genes selected to be potentially active in the mosquito germline of both males and females was tested for 9 independent isolines by crossing the Cas9 strain to the previously characterised kmosgRNA strain and monitoring the inheritance of the transgenes in the F2 progeny considering also the parental effect. The authors identified one cas9 construct to efficiently bias the inheritance in both males and females and explored with a mathematical model the potential for invasion compared to previously published constructs. The data generated are very clear and the experimental design neat to uncover the inheritance rate of the constructs. The manuscript could be improved in terms of clarity, especially description of results about grandparents contributions, which is very difficult

to read. I find the methods very poor of details, for example about the construct design, the calculation of fitness data, release scenarios for the population model.

Overall I think the data are of interest for the gene drive community and worth to be published in Nature Comms, but I ask the authors to revise the manuscript to improve its clarity, expand the discussion for the points that are relevant (see below), expand the methodology to provide more details. I think some additional data (eg target site resistance) could benefit the manuscript but I leave it to the authors' judgment the need to include them.

Specific comments below:

- The choice of promoters is not novel at all. *nanos* and *zpg* are widely used in *Anopheles* and *Drosophila* gene drives and I struggle to consider them 'novel'. It was a very obvious choice. I would suggest to remove the word novel referred to the choice of the regulatory sequences.
- Add line numbers for facilitating the review.
- Include a schematic of the construct or a detailed description of the molecular components in the methods. For instance, it is not indicated the presence of the fluorescent markers.
- Number of lines for each construct is not clearly outlined.
- Why there are two different *zpg*-Cas9 in Table S1? And Suppl Table 1 is lacking info for *sds3*-Cas9
- Fig 1 legend (D): replace *nosA1*-Cas9 with *nosA2*-Cas9
- No indication of number of replicates or population size for the drive assessment assay.
- Fig 1 D legend: "The integration sites for *EwaldC1*-Cas9, *EwaldC2*-Cas9, *nosA1*-Cas9, and *sds3E1*-Cas9 isolines are likely linked to the *kmo* locus indicated by the low meiotic recombination between these Cas9 and *kmosgRNAs* elements." If the integration site of *EwaldC1*-Cas9, *EwaldC2*-Cas9, *nosA2*-Cas9, and *sds3E1*-Cas9 isolines are linked to the *kmo* locus you would expect both constructs to segregate together. It looks instead that there is a lack of Cas9/*kmosgRNA* that could indicate instead a fitness cost.
- "We found grandparental sex at the P0/M0 levels to significantly (Fisher's exact test, $p < 0.0016$) affect the ability of the P1/M1 trans-heterozygotes to cause inheritance bias by comparing the P1 males and females to their M1 equivalents. While an identical trend was observed between P1 and M1 males (Fisher's exact test, $p < 0.0001$) from the *sds3G1*-Cas9 crosses, P0/M0 sex did not appear to have caused any significant differences to the biasing efficiency between the P1/M1 females (Fisher's exact test, $p = 0.53$)." I find this sentence confusing and contradicting: By comparing Fig 1C and 2B it seems that there is no effect from grandparent sex (parental contribution) for *sds3G1*-Cas9 crosses, while there is for *shuC1*-Cas9, *shuB1*-Cas9 and *shuC2*-Cas9. Please rephrase. The whole section is very difficult to read.
- Pag 17: "Inheritance rates of *kmosgRNAs* from trans-heterozygous males and females were observed to be 68% and 77%, respectively." While few lines after it reads "The average inheritance rates of *kmosgRNAs* from trans-heterozygous males and females crossed to WT mosquitoes are found to be ~86% ($n = 3,937$ larvae) and ~94% ($n = 1,038$ larvae), respectively". No clear where the former data are coming from, since it seems to provide different transgenic rate values.
- Pag 20 "Given that fitness of the *kmosgRNAs* element could be potentially improved by providing a recoded *rescue37*, we go on to model a hypothetical scenario in which the *kmosgRNAs* element has reduced fitness costs, in line with those for *wU6b-GDe* (Fig 4)". This could be true, but there is a fitness cost also associated with the *sds3G1* line that cannot be reduced by recoding the insertion site, considering its efficacy is also strongly impacted by positional effect (see no homing for the other *sds3* isolines). This point needs to be expanded.
- Pag 21: "The lack of any indication of Cas9 activities from *Ewald*-Cas9 and *zpg*-Cas9 in the present study may be due to the putative regulatory elements not encompassing the regions necessary for transcription and/or translation of Cas9." Given that *zpg* is highly effective in driving expression of Cas9 for gene drives in *Anopheles*, this is very surprising. I understand and share the idea to test a longer version of the promoter (4kb) which surprisingly still did not bias inheritance. Did the author investigated bioinformatically the promoter region they selected to identify if any putative transcription factor binding sites are present? A rt-PCR on the Cas9 would also shed light if the activity and homing rate between the lines is correlated somehow to Cas9 expression.
- The authors little focus on the potential to resistance (only half sentence on line 8 of pag 23) due to the deposition effect of the *shu* and *sds3* regulatory regions, especially where high degree of white-eye mosaicism was observed. An analysis of the repair outcomes profiles (for instance by

amplicon sequencing of the target site) could help both to understand if the low transgenic rate observed is due to low cleavage activity or by NHEJ repair, which can impact the dynamics of invasion of the split drive and contribute to generation of resistance.

- The *sds3* promoter seems to not be consistently effective in biasing the inheritance but there is a strong positional effect (see the difference between *sd3A1-E1* and *G1* isolines). This is briefly mentioned in the discussion, but I would like the authors to discuss further this result and its implication.
- I agree with the conclusion of the authors that the data presented showed that *Ae. aegypti* is not recalcitrant to Cas9-based gene drives, and I share the optimistic view that fitness improvements on the current design and components is feasible. However, I would like the authors to be more cautious in the conclusion that the combination of the elements they propose (*sds3G1-Cas9;kmosgRNAs*) can be considered an effective gene drive. In order to say that I think a population cage trial would be needed to validate the modelling prediction, which are promising but deemed by many assumptions, first of all about the fitness of the genotype, impact of target site resistance, and the synergic effect of the two split components on the fitness
- The supplementary data file 1 needs description of the content.

Point-by-point response to reviewers

Please find our responses below the reviewer comments, in italics for clarity.

Reviewer #1 (Remarks to the Author):

Nature Communications Review:

In their manuscript entitled “Novel germline regulatory elements enable highly efficient gene drive in the yellow fever mosquito *Aedes aegypti*,” Anderson and colleagues describe novel transgenic strains of *Aedes aegypti* that utilize germline regulatory elements to drive Cas9 expression. They demonstrate the efficacy of these lines to bias the inheritance of a kmo-sgRNA construct as a proof-of-principle for an efficient gene drive of the ‘split drive’ style.

Gene drives (and CRISPR/Cas9 followed by HDR in general) in *Aedes aegypti* has until now under-performed as compared to *Anopheles gambiae*. With these new lines, Anderson and colleagues have helped to close that gap and their work will be of substantial interest to a wide variety of mosquito biologists, including those interested in developing gene drives for vector control as well as those using CRISPR/Cas9 and HDR to understand basic mosquito biology. In addition to establishing these new lines as viable drivers for gene drive purposes, they use carefully tracked crossing experiments to demonstrate convincingly that maternal inheritance of Cas9 can contribute functionally to inheritance biasing in these schemes. Finally, having settled on one line (SDS3G1-Cas9) that had high levels of inheritance bias in multiple crossing schemes, they evaluated crosses derived from individual animals to determine the levels of inter-individual variability. In all, these data are convincing and experiments were well controlled (but see comments below).

Major comments:

1) There is a marked difference in activity among different isolines. In addition to the linkage to the m locus for one isolate of sds3, is there any evidence of where the other insertions lie? Approaches such as splinkerette PCR or TagMapping can be used to map piggyBac insertions and knowing where these lines lie could be very useful for future work in mosquito transgenesis. This is especially important as the point raised in the discussion acknowledges: “that integration sites... played a significant role in transgene expression”

Adapter-ligation mediated PCR was performed to identify the insertion sites of the shu-Cas9 and sds3-Cas9 lines. These sites are listed in Supplementary Table S5. There was no apparent pattern between lines that seemed to express and those that did not, with insertions for both in intergenic and intronic regions. This has now been expanded in the discussion.

Minor comments:

2) I found Figure 1d and 2c very difficult to interpret at first, especially the differences between the WT and not scored categories in the top two rows. Could a pattern be used instead of different border colours?

The borders for these categories are now drawn using dashes lines.

3) The graphic design of Figure 2 was initially confusing to me as I mistook panel 2b as referring to the left arm of the cross depicted in 2a (and vice-versa for 2c). These panels could be potentially re-arranged to avoid that confusion?

The panels have now been rearranged.

Reviewer #2 (Remarks to the Author):

The manuscript “Novel germline regulatory elements enable highly efficient gene drive in the yellow fever mosquito *Aedes aegypti*” by Anderson and colleagues describes the ability of five different regulatory sequences to express Cas9 and bias the inheritance of the gRNA construct in a context of split gene drive in *Aedes aegypti*. The promoter and 3’UTR of those five genes selected to be potentially active in the mosquito germline of both males and females was tested for 9 independent isolines by crossing the Cas9 strain to the previously characterised kmosgRNA strain and monitoring the inheritance of the transgenes in the F2 progeny considering also the parental effect. The authors identified one cas9 construct to efficiently bias the inheritance in both males and females and explored with a mathematical model the potential for invasion compared to previously published constructs. The data generated are very clear and the experimental design neat to uncover the inheritance rate of the constructs. The manuscript could be improved in terms of clarity, especially description of results about grandparents contributions, which is very difficult to read. I find the methods very poor of details, for example about the construct design, the calculation of fitness data, release scenarios for the population model.

We would like to thank the reviewer for their time and feedback. We have now rewritten the section about grandparental contributions. We have also provided more details to the methods regarding construct design as well as created a diagram of the constructs, which is Supplementary Figure S1 and deposited the full plasmid sequences to NCBI. The explanation for the fitness data calculations was previously in the Supplementary S1 File. The information from this file has now been incorporated into the Supplementary Text 1, detailing the model. We hope this is now more available to the reader.

Overall I think the data are of interest for the gene drive community and worth to be published in Nature Comms, but I ask the authors to revise the manuscript to improve its clarity, expand the discussion for the points that are relevant (see below), expand the methodology to provide more details. I think some additional data (eg target site resistance) could benefit the manuscript but I leave it to the authors’ judgment the need to include them.

Specific comments below:

- The choice of promoters is not novel at all. nanos and zpg are widely used in Anopheles and Drosophila gene drives and I struggle to consider them ‘novel’. It was a very obvious choice. I would suggest to remove the word novel referred to the choice of the regulatory sequences.

We thank the reviewer for their insightful comments, the use of the term novel has been removed.

- Add line numbers for facilitating the review.

Line numbers have been added.

- Include a schematic of the construct or a detailed description of the molecular components in the methods. For instance, it is not indicated the presence of the fluorescent markers.

A new figure has been added to the Supplementary file (Figure S1) depicting the transgene constructs.

- Number of lines for each construct is not clearly outlined.

Details for the number of lines generated as well as the specific isolines used in this study have been added to Supplementary Table S1.

- Why there are two different zpg-Cas9 in Table S1? And Suppl Table 1 is lacking info for sds3-Cas9

The two zpg lines mentioned in the table are the two different zpg promoter fragments tested, this has been clarified. Details for sds3 have been added.

- Fig 1 legend (D): replace nosA1-Cas9 with nosA2-Cas9

This has been corrected.

- No indication of number of replicates or population size for the drive assessment assay.

Sample sizes are now included in Figures 1-3.

- Fig 1 D legend: "The integration sites for EwaldC1-Cas9, EwaldC2-Cas9, nosA1-Cas9, and sds3E1-Cas9 isolines are likely linked to the kmo locus indicated by the low meiotic recombination between these Cas9 and kmosgRNAs elements." If the integration site of EwaldC1-Cas9, EwaldC2-Cas9, nosA2-Cas9, and sds3E1-Cas9 isolines are linked to the kmo locus you would expect both constructs to segregate together. It looks instead that there is a lack of Cas9/kmosgRNA that could indicate instead a fitness cost.

We have now clarified that we meant linkage in trans in the initial version of our manuscript. We hypothesise that the piggybac based random integrations of the Cas9 isolines mentioned has occurred near the kmo gene (not the kmo^{sgRNAs} insertion). When they were crossed to each other to generate P₁ trans-heterozygotes, the two transgenes become linked in trans (i.e. on opposite chromosomes). In this case, due to low recombination rates between the two transgenes, the trans-heterozygote and non-transgenic genotypes comprise a lower proportion of the P₂ compared to the other two genotypes, as was observed. We do not expect this to be a fitness cost to trans-heterozygotes because this would not explain the low representation of non-transgenic genotype in P₂.

- “We found grandparental sex at the P0/M0 levels to significantly (Fisher’s exact test, $p < 0.0016$) affect the ability of the P1/M1 trans-heterozygotes to cause inheritance bias by comparing the P1 males and females to their M1 equivalents. While an identical trend was observed between P1 and M1 males (Fisher’s exact test, $p < 0.0001$) from the sds3G1-Cas9 crosses, P0/M0 sex did not appear to have caused any significant differences to the biasing efficiency between the P1/M1 females (Fisher’s exact test, $p = 0.53$).” I find this sentence confusing and contradicting: By comparing Fig 1C and 2B it seems that there is no effect from grandparent sex (parental contribution) for sds3G1-Cas9 crosses, while there is for shuC1-Cas9, shuB1-Cas9 and shuC2-Cas9. Please rephrase. The whole section is very difficult to read.

This section has been entirely rewritten.

- Pag 17: “Inheritance rates of kmosgRNAs from trans-heterozygous males and females were observed to be 68% and 77%, respectively.” While few lines after it reads “The average inheritance rates of kmosgRNAs from trans-heterozygous males and females crossed to WT mosquitoes are found to be ~86% ($n = 3,937$ larvae) and ~94% ($n = 1,038$ larvae), respectively”. No clear where the former data are coming from, since it seems to provide different transgenic rate values.

The first sentence refers to bgcn-Cas9 which is described in the sentence prior to the one the reviewer mentions. Later in the same paragraph we are describing the data for another promoter, sds3. We have clarified this section.

- Pag 20 “Given that fitness of the kmosgRNAs element could be potentially improved by providing a recoded rescue37, we go on to model a hypothetical scenario in which the kmosgRNAs element has reduced fitness costs, in line with those for wU6b-GDe (Fig 4).”. This could be true, but there is a fitness cost also associated with the sds3G1 line that cannot be reduced by recoding the insertion site, considering its efficacy is also strongly impacted by positional effect (see no homing for the other sds3 isolines). This point needs to be expanded.

While the reviewer is correct there is a fitness cost associated with the sds3G1-Cas9 line, this cost is not high enough to impede the drive if the fitness cost due to the kmo^{sgRNAs} is improved, as can be seen in Figure 4, the blue line with dots. The positional effect on sds3 expression has been further expanded in the discussion, and the implications have been toned down.

- Pag 21: “The lack of any indication of Cas9 activities from Ewald-Cas9 and zpg-Cas9 in the present study may be due to the putative regulatory elements not encompassing the regions necessary for transcription and/or translation of Cas9.” Given that zpg is highly effective in driving expression of Cas9 for gene drives in Anopheles, this is very surprising. I understand and share the idea to test a longer version of the promoter (4kb) which surprisingly still did not bias inheritance. Did the author investigated bioinformatically the promoter region they selected to identify if any putative transcription factor binding sites are present? A rt-PCR on the Cas9 would also shed light if the activity and homing rate between the lines is correlated somehow to Cas9 expression.

A scan with TFsitiescan revealed three *D. melanogaster* core promoter motifs within the 1kb immediately upstream of the TSS. A recent (Oct 17, 2022) paper has also used the *Ae. aegypti* zpg regulatory regions for expressing Cas9, in their case in a single locus design. They use the HWE strain of *Ae. aegypti* and ~1.7kb upstream as the promoter, and ~1.3kb downstream as the 3'UTR. They note a 144bp deletion 825-bp upstream of the +1 ATG of the zpg ORF (corresponding to positions AaegL5_2:84863229–848632372). They observed a very modest inheritance bias in females (66%), and none in males (56%) leading them to suggest a sex-specific effect on Cas9 expression, also due to position effect. This new reference has been added and this is now brought up in the discussion.

Reid W, Williams AE, Sanchez-Vargas I, Lin J, Juncu R, Olson KE, Franz AWE. Assessing single-locus CRISPR/Cas9-based gene drive variants in the mosquito *Aedes aegypti* via single generation crosses and modeling. *G3 (Bethesda)*. 2022 Oct 17:jkac280. doi: 10.1093/g3journal/jkac280. Epub ahead of print. PMID: 36250791.

- The authors little focus on the potential to resistance (only half sentence on line 8 of pag 23) due to the deposition effect of the shu and sds3 regulatory regions, especially where high degree of white-eye mosaicism was observed. An analysis of the repair outcomes profiles (for instance by amplicon sequencing of the target site) could help both to understand if the low transgenic rate observed is due to low cleavage activity or by NHEJ repair, which can impact the dynamics of invasion of the split drive and contribute to generation of resistance.

We have added more to the discussion related to resistance due to deposition. It should be noted that we found the highest inheritance bias with sds3 when the Cas9 was inherited maternally, this is the reasoning behind the M experimental crosses performed in Fig 2. We have now more explicitly mentioned this and expanded the discussion. Due to the multiplexed nature of the *kmo^{sgRNAs}* line, analysis of target site resistance is not within the scope of this manuscript, but we have addressed that in a separate paper currently under review. It is also available now as a preprint (<https://www.biorxiv.org/content/10.1101/2022.08.12.503466v1.full.pdf>).

- The sds3 promoter seems to not be consistently effective in biasing the inheritance but there is a strong positional effect (see the difference between sd3A1-E1 and G1 isolines). This is briefly mentioned in the discussion, but I would like the authors to discuss further this result and its implication.

The discussion regarding the positional effect has been significantly expanded.

- I agree with the conclusion of the authors that the data presented showed that *Ae. aegypti* is not recalcitrant to Cas9-based gene drives, and I share the optimistic view that fitness improvements on the current design and components is feasible. However, I would like the authors to be more cautious in the conclusion that the combination of the elements they propose (sds3G1-Cas9;kmosgRNAs) can be considered an effective gene drive. In order to say that I think a population cage trial would be needed to validate the modelling prediction, which are promising but deemed by many assumptions, first of all about the fitness of the genotype, impact of target site resistance, and the synergic effect of the two split components on the fitness

This statement has been toned down in the discussion.

- The supplementary data file 1 needs description of the content.

This has been added.

Reviewers' Comments:

Reviewer #1:

Remarks to the Author:

In this revision, Anderson, Gonzalez, Ang et al. have provided thoughtful and thorough responses to both reviewers' comments. I especially appreciated the revised and expanded discussion of positional effects on the efficacy of transgenic components in this type of system and, in short, I am satisfied with the additional data and qualifying statements in the revised manuscript and recommend that it be published in the current state.

Reviewer #2:

Remarks to the Author:

I thank the authors to take in consideration all the comments I had on the manuscript and to largely improve the manuscript. The discussion and the description of results is now more complete and easier to read. I appreciate the updated methods and supplementary file with additional information. I also like better the title. I am happy to accept the manuscript for publication.